# Evaluation of *Aedes aegypti* control intervention with pyriproxyfen by lcWGS in Manacapuru, Amazonas, Brazil

**Lorena Ferreira de Oliveira Leles**[1,2]*, **Marcus Vinicius Niz Alvarez**[3], **Jose Joaquin Carvajal Cortés**[1], **Diego Peres Alonso**[3], **Paulo Eduardo Martins Ribolla**[3], **Sérgio Luiz Bessa Luz**[1,2]

**1** Laboratório de Ecologia de Doenças Transmissíveis na Amazônia, Instituto Leônidas e Maria Deane—Fiocruz Amazônia, Manaus, Brasil, **2** Programa de Pós-Graduação em Biologia Parasitária, Instituto Oswaldo Cruz (IOC), Rio de Janeiro, Brasil, **3** Universidade do Estado de São Paulo (UNESP)—Instituto de Biotecnologia e Biociências, Botucatu, Brasil

* lorenaleless@gmail.com

**Data Availability Statement:** The authors confirm that all data underlying the findings are fully available without restriction. Data is available at

## Abstract

### Background

*Ae. aegypti* mosquitoes are considered a global threat to public health due to its ability to transmit arboviruses such as yellow fever, dengue, Zika and Chikungunya to humans. The lack of effective arboviral vaccines and etiological treatments make vector control strategies fundamental in interrupting the transmission cycle of these pathogens. This study evaluated *Ae. aegypti* mosquito populations pre- and post-intervention period with disseminating stations of the larvicide pyriproxyfen to understand its potential influence on the genetic structure and population diversity of these vectors.

### Methodology/Principal findings

This study was conducted in Manacapuru city, Amazonas, Brazil, where 1,000 pyriproxyfen dissemination stations were deployed and monitored from FEB/2014 to FEB/2015 (pre-intervention) and AUG/2015 to JAN/2016 (post-intervention). Low-coverage whole genome sequencing of 36 individuals was performed, revealing significant stratification between pre- and post-intervention groups (pairwise $F_{ST}$ estimate of 0.1126; p-value < 0.033). Tajima's D estimates were -3.25 and -3.07 (both p-value < 0.01) for pre- and post-intervention groups, respectively. Molecular diversity estimates (Theta(S) and Theta(Pi)) also showed divergences between pre- and post-intervention groups. PCA and K-means analysis showed clustering for SNP frequency matrix and SNP genotype matrix, respectively, being both mainly represented by the first principal component. PCA and K-means clustering also showed significant results that corroborate the impact of pyriproxyfen intervention on genetic structure populations of *Ae. aegypti* mosquitoes.

### Conclusions/Significance

The results revealed a bottleneck effect and reduced mosquito populations during intervention, followed by reintroduction from adjacent and unaffected populations by this vector. We

NCBI with the following BIOPROJECT number PRJNA970519.

**Funding:** The author(s) received no specific funding for this work.

**Competing interests:** The authors have declared that no competing interests exist.

highlighted that low-coverage whole genome sequencing can contribute to genetic and structure population data, and also generate important information to aid in genomic and epidemiological surveillance.

## Author summary

The *Aedes aegypti* mosquito is a global public health threat, transmitting diseases like yellow fever, dengue, zika and Chikungunya, requiring several vector control strategies. Traditional methods such as active search and treatment of accessible breeding sites by control agents can be supplemented using PPF dissemination stations. The tactic uses adult mosquitoes themselves to disseminate the larvicide adhered to their bodies, between treated and untreated breeding sites, which are usually cryptic and inaccessible. This tactic can easily be incorporated into integrated vector management strategies, reducing the emergence of adult mosquitoes and, consequently, arboviruses transmission. However, its impact on *Ae. aegypti* genetic and population structure remain unknown. The PPF intervention caused a bottleneck in the genetic structure, an event in which a population undergoes a drastic reduction in population density, in this case due to the pressure caused by the larvicide, followed by reintroduction of adult mosquitoes from nearby areas not exposed to PPF. This study uses a cost-effective low-coverage whole genome sequencing method to analyze *Ae. aegypti* genetic population structure based on mitochondrial genome, comparing pre- and post-intervention periods with pyriproxyfen. This approach can reveal both short and long-term effects of this control strategy in the vector genome dynamics. These findings have direct implications for planning control activities, enhancing genomic surveillance and epidemiological surveillance, contributing to global efforts against vector-borne diseases.

## Introduction

Currently *Ae. aegypti* is considered a global threat to public health not only due to its aggressiveness (i.e. discomfort caused by the bites) and, invasiveness (i.e. occurrence in wild and urban areas), but also to its ability to transmit arboviruses to humans, such as yellow fever, dengue, Zika and Chikungunya [1,2]. Recently, the Zika virus outbreak and its association with serious damage to the nervous system, including microcephaly, have resulted in national and international public health emergencies [3].

The lack of effective arboviral vaccines (except for yellow fever) and etiological treatments make vector control strategies crucial to interrupt the transmission cycle of these pathogens, which has been shown effective against other vector-borne diseases, such as malaria [4]. In fact, the World Health Organization (WHO) estimates that in the next fifteen years, about 50–60% of investment funds for controlling and eliminating neglected tropical diseases will be allocated to developing new insecticides and transmission control strategies. Understanding the impact of these strategies on vector genomes and exploring cost-effective alternatives is essential, because this approach enables active participation of developing countries, where most of these diseases are prevalent, in the control and elimination efforts [5].

Traditional methods for controlling *Ae. aegypti* include the use of adulticides, active search, and treatment of accessible breeding sites by health control agents [6,7]. Despite the benefits obtained using these methods, one of the main challenges faced in these strategies is the low

efficiency of breeding sites discovery and treatment. Since *Ae. aegypti* oviposition characteristically occurs in small, cryptic, and inaccessible breeding sites, these can go unnoticed during vector control campaigns involving insecticide application [7,8].

An effective strategy to increase the number of treated breeding sites leverages the dissemination of potent larvicides such as pyriproxyfen (PPF), by the mosquitoes themselves, into difficult-to-access breeding sites, interfering in larvae and pupae growth, preventing the vector development [9,10]. The use of this larvicide in disseminating stations (DSs) has proven to effectively reduce the transmission of arboviruses due to the spread of the larvicide by the adult female mosquito during oviposition in untreated breeding sites, consequently resulting in a smaller number of competent vectors available in the environment [4,8]. Despite being effective, the impact of this control strategy on the biological dynamics, genetic and population structure of these vectors remains unknown [8,11].

For *Ae. aegypti*, few studies correlated genetic and population structures with robust molecular techniques and a rigorous epidemiological database and most of these studies identify genetic variations, their distribution and frequency among populations using microsatellite markers and primer-specific SNP genotyping [2,12–14]. Recent advancements in new sequencing techniques, such as *low-coverage whole genome sequence* (lcWGS), have provided extensive data that enable a better understanding of structural dynamics and population genetics of several organisms. Therefore, these approaches allow the evaluation of different control strategies by identifying significant microvariations in populations of important vectors, also allowing genome assembly, gene panels construction and search for SNPs of interest, as already observed for *Anopheles darlingi* [15].

These new sequencing techniques brought significantly reduced costs in population genetics studies, but genomic sequencing expenses remain high for population genetics studies due the large number of samples that need to be sequenced [16]. A viable alternative for large-scale genomic sequencing is the lcWGS, which allows obtaining sufficient genomic data for understanding population genetic structure, associating high reliability and low cost [16,17]. Despite the relatively large genome size of *Ae. aegypti*, lcWGS can still effectively recover mitochondrial genome markers, even when the average depth of nuclear genome sequencing is limited [18,19].

In this paper, we applied a cost-effective, *low-coverage whole genome sequencing* method to evaluate *Ae. aegypti* mosquito populations during the pre- and post-intervention periods with disseminating stations of the larvicide pyriproxyfen [11,16]. This approach can help us understand its potential influence on the genetic structure and population diversity of these vectors, based on long-term exposure to PPF. These findings have direct implications for planning control activities, enhancing genomic and epidemiological surveillance, contributing to global efforts against vector-borne diseases [17,20].

## Material and methods

### Ethics approval and consent to participate

Dr. Sérgio Luiz Bessa Luz is permanently licensed (27733–1) by Brazilian Institute of Environment and Natural Resources (IBAMA) to collect disease vectors. All procedures were conducted with residents' permission. Formal approval was not required for mosquito collection in urban environments. Ethical review and approval were waived for this study.

### Study area and mosquito surveillance

Samples used in this paper are from a previously developed project titled *"Insecticide Spread by Mosquitoes for Vector Control and Potential Blocking of Arbovirus Epidemics"*[8]. In this study, we conducted a previous intervention study using DSs contaminated with pyriproxyfen in the

Amazonas State. Collections were carried out in the municipality of Manacapuru (Fig 1), Central Amazon, Brazil, a city with approximately 60,000 inhabitants and 13,500 dwellings (data from Brazilian Institute of Geography and Statistics–IBGE).

One hundred dwellings were randomly selected to produce the maximum coverage of the study area. One thousand pyriproxyfen dissemination stations were placed across the city, consisting of black plastic cups and water to attract *Aedes* mosquitoes. DSs were verified twice

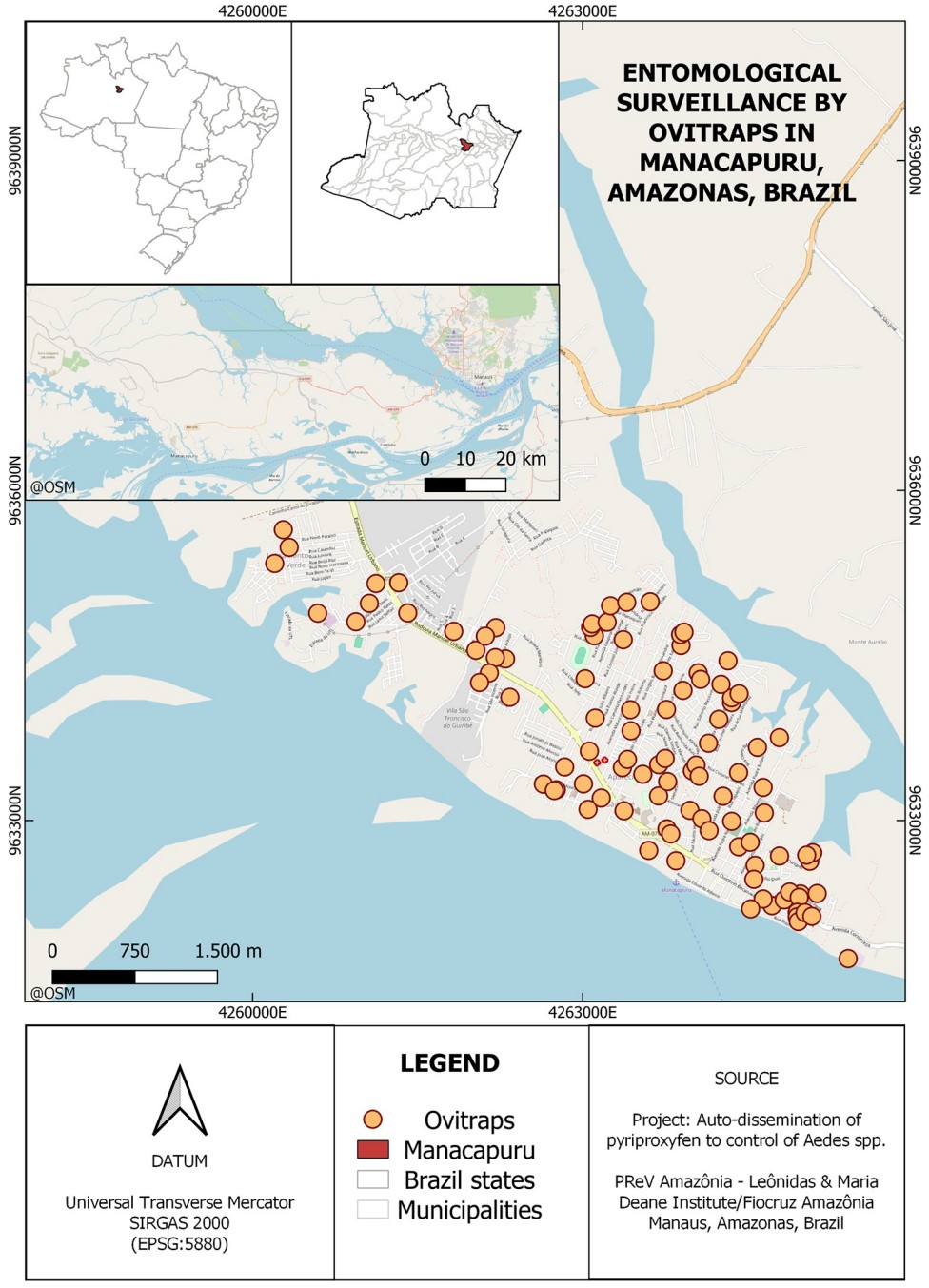

**Fig 1. Study site: The city of Manacapuru, Amazonas, Brazil.** The orange circles indicate the location of the dwellings monitored for mosquito vectors. The map was drawn using the maps library on software QGIS (www.ibge.gov.br/geociencias/organizacao-do-territorio/malhas-territoriais/15774-malhas.html).

a month from FEB/2014 to FEB/2015 (pre-intervention period), MAR/2015 to JUL/2015 (intervention) and AUG/2015 to JAN/2016 (post-intervention period) as shown in Fig 2.

The juvenile specimens of *Ae. aegypti* were collected, morphologically identified and later emerging in adult mosquitoes in laboratory conditions. The samples collected were stored in 100% alcohol and placed in a freezer (4˚C). Specimens were selected by pre- and post-intervention period, also considering rainy and dry seasonal periods.

## Sample preparation and sequencing

For DNA extraction, each specimen was extracted individually using a DNeasy Blood & Tissue (Qiagen, USA) and DNA quantification was performed by fluorometric quantitation using a Thermo Fisher Scientific QuBit dsDNA HS Assay Kit, both according to the manufacturer's recommendations.

DNA libraries were prepared using one-fifth of the total recommended volume for the Nextera XT Library prep kit (Illumina) according to the manufacturer's recommendations. DNA samples were multiplexed, loaded on a mid-output flow cell and sequenced using the NextSeq500 platform (Illumina) in a 151-cycle single-read run. Sequence-quality analyses were performed using the FASTQC program [21] and reads were used if results from all analysis modules were approved without errors.

## Species identification

Sequencing data was aligned with the *Ae. aegypti* cytochrome oxidase subunit I (COI) reference sequence (available at KC913582.1) using Burrows-Wheeler Aligner (BWA) software [22]. After alignment, the individual COI consensus sequences were generated using the SamTools software package [23]. The BLASTn tool was used for multi-species identification using the individual generated COI consensus sequence [24]. Only the highest matching result from BLAST was used. Specimens were discarded if e-value $> 1e^{-100}$, identities $< 200$, and identity $< 90\%$, and if the matching sequence was not identified as *Ae. aegypti*.

## Variant calling

Sequencing data quality analysis was performed using FASTQC [21] and reads filtering, trimming and adapter removal procedures were performed using Trimmomatic [25], dropping

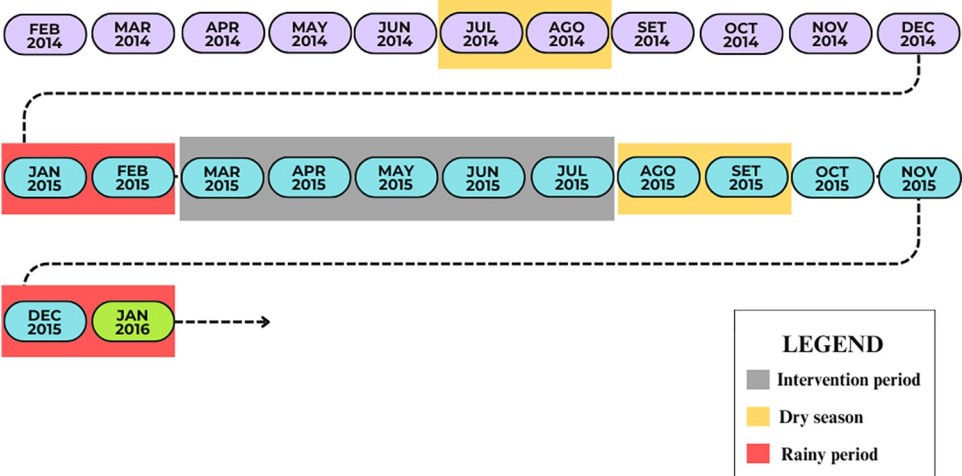

**Fig 2.** Timeline of collection period, highlighted intervention period (gray) and sampling at dry (yellow) and rainy (red) seasons.

reads with length less than 80 base pairs. All sequencing reads were aligned against the reference mitochondrial genome of *Ae. aegypti* (NCBI NC_035159.1, available from GenBank) using the BWA program [22]. Variant calling was performed with the SamTools software package [23], and the panel of variants was exported in the VCF version 4.2 format. SNPs were removed from the variant panel based on a minimum allele frequency (MAF) < 0.1 and missing data (MD) > 0.5 using the LCVCFtools program [26]. Genotypes with sequencing depth (DP) < 5 or phred genotype-quality score (GQ) < 20 were filtered.

### Mitochondrial population genetics inferences

Statistical analyses were performed with PLINK 1.9 [27] and graphs and figures were generated using the GGPlot2 package in RStudio [28]. Pairwise $F_{ST}$, nucleotide diversity ($\theta_S$ and $\theta_\pi$) and Tajima's D were estimated using ARLEQUIN 3.5 [29]. The principal component analysis (PCA) procedure was calculated by using both the SNP frequencies estimated for each group and pairwise genetic distance IBS (Identity-by-state matrix). Hierarchical clustering analysis was performed for both PCA results using pvclust v2.2–0 package for R [30].

## Results

### Data collection and DNA extraction

During the monthly checks from 2014 to 2016, 1.933 specimens of *Ae. aegypti* were collected and morphologically identified, being 455 samples from the dry seasonal period and 1.478 samples from the rainy seasonal period. A total of 36 individuals were submitted to low coverage sequencing, followed by species identification: 18 individuals from the pre-intervention and 16 individuals from the post-intervention period, with half of the samples corresponding to the dry and rainy season from each intervention group. All individuals were identified as *Ae. aegypti*.

### Low-coverage whole genome sequencing (lcWGS) performance

A total of 6.705 reads were properly mapped with the mitogenome reference. The sequencing GC content was 21.03%. The sequencing depth (Fig 3) and mapping coverage distribution (Fig 4) are represented below. The average sequencing depth was approximately 1.43 ± 0.26. After variant calling, the final dataset consisted of 97 SNPs distributed throughout the mitogenome. The SNP average sequencing depth and quality was 4.00 ± 0.71 and 35.9 ± 3.73, respectively. The SNP density was 5.7 SNP/Kbp.

### Statistical and structural analysis

Significant stratification was observed between pre-intervention and post-intervention groups, with a pairwise $F_{ST}$ estimate of 0.1126 (p-value < 0.033, number of permutations = 1000). The estimates of Tajima's D of pre-intervention and post-intervention groups were -3.25 (p-value < 0.01) and -3.07 (p-value < 0.01) respectively. Theta(S) molecular diversity estimates for the pre-intervention and post-intervention groups were 24.7 ± 8.8 and 10.2 ± 3.9, respectively. Theta(Pi) molecular diversity estimates for pre-intervention and post-intervention groups were 5.5 ± 3.1 and 2.9 ± 1.6, respectively.

The PCA analysis showed clustering for SNP frequency matrix, as the figure below (Fig 5). The clustering is mainly represented by the first principal component which corresponds to at least half of the total variance by the model. The SNP rotation estimates were relatively homogeneous. The k-means clustering also showed significant clusters (Fig 6), given the approximately unbiased p-value estimates > 95% (alpha < 0.05).

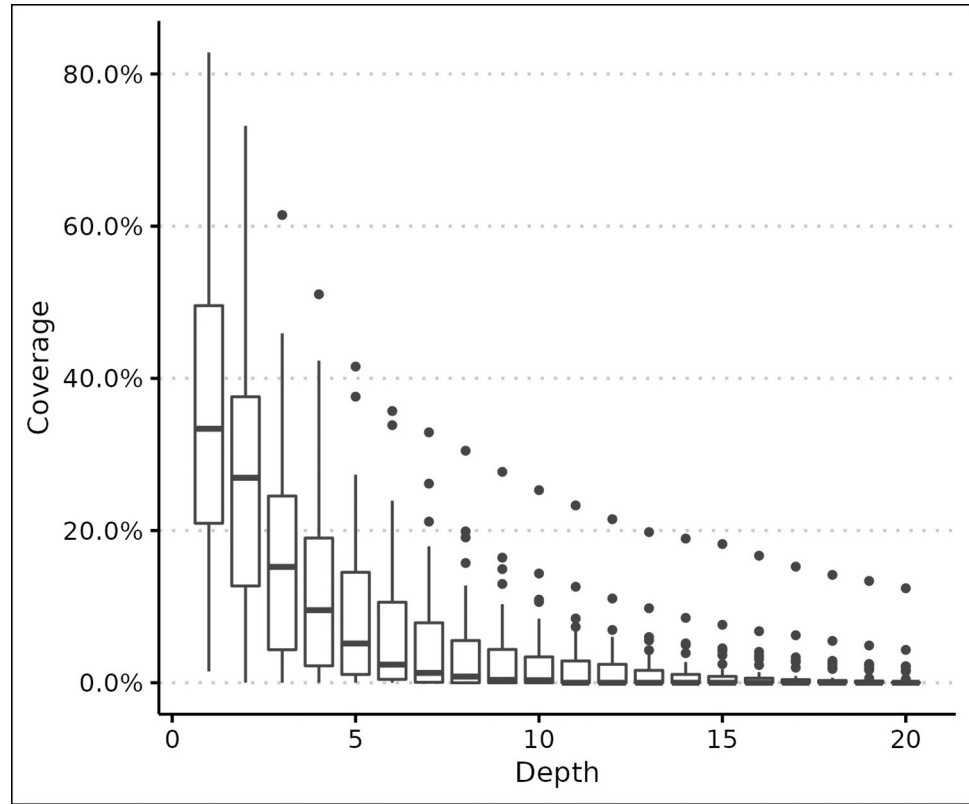

**Fig 3. Mitogenome sequencing coverage boxplot diagram given the respective depth.**

The PCA analysis showed clustering for SNP genotype matrix, as the figure below (Fig 7). The clustering is mainly represented by the first principal component which corresponds to the major explained variance by the model.

## Discussion

This study provides a comprehensive analysis of the genetic structure of field-collected *Ae. aegypti* in Manacapuru, Amazonas, Brazil, before and after intervention involving disseminating stations of pyriproxyfen. From an epidemiological perspective, the intervention led to a significant decrease in juvenile catch, from an average of 3.20 individuals per sentinel/month before the intervention to less than one juvenile during dissemination. Moreover, the initial monthly mortality rate of 6.6% for *Ae. aegypti* increased dramatically to an average of 96% during the intervention period [8]. The combination of reduced juvenile capture due to the near absence of mosquitoes population, and heightened juvenile mortality resulting from exposure to PPF, led to a significant decline in adult mosquito emergence during the intervention period. The effect was particularly drastic, considering the nearly complete absence of mosquitoes due to the intervention [7,8].

Based on Tajima's D estimate, there is indicative evidence of a population expansion, potentially following a recent bottleneck effect triggered by the application of PPF. Stratification analysis ($F_{ST}$) reveals moderate stratification between pre- and post-intervention groups. Interestingly, both groups presented very similar signs of stratification, considering both the level of stratification and number of permutations. To our knowledge, there are currently no

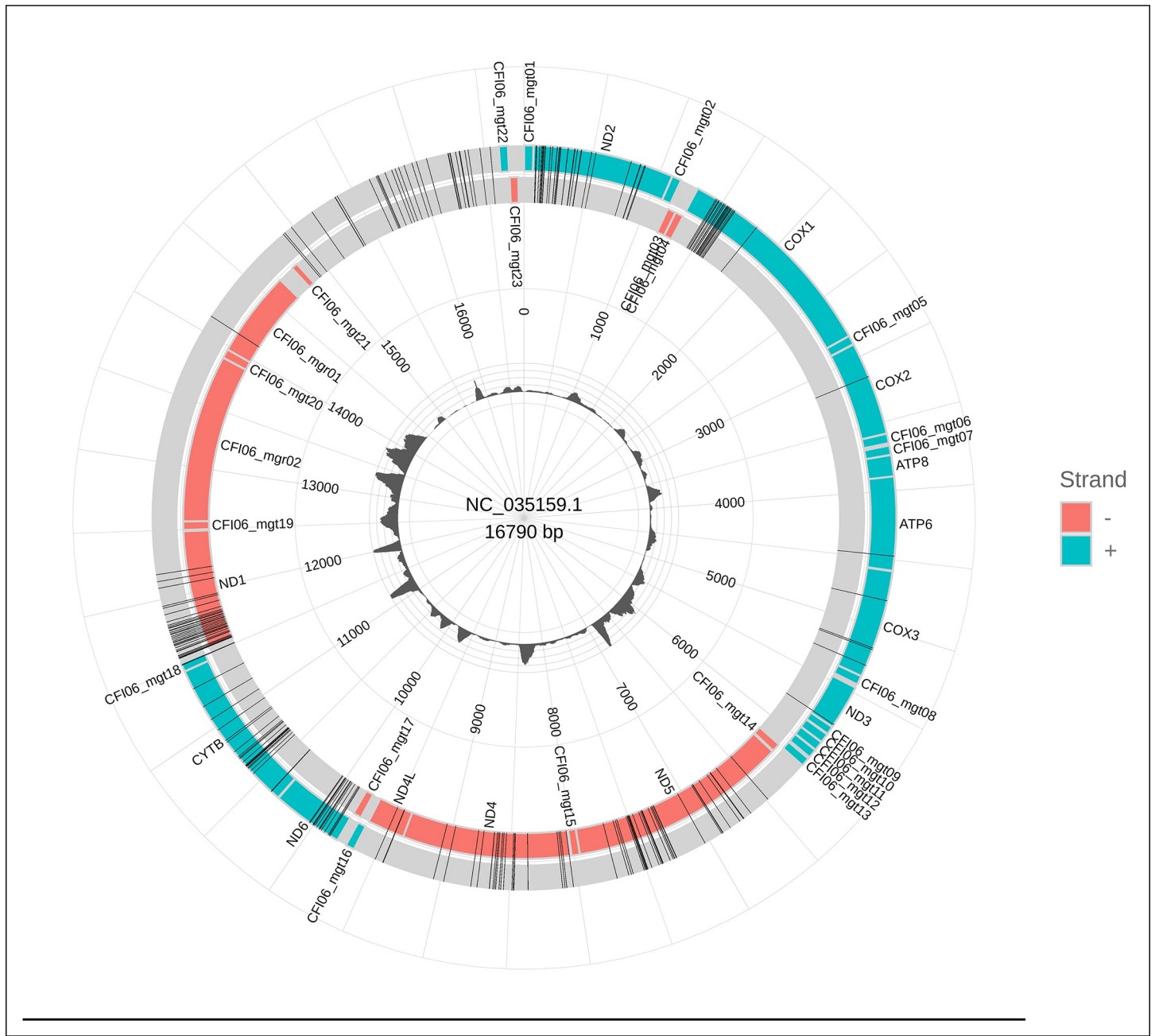

**Fig 4. The reference mitogenome.** SNPs are highlighted as black vertical lines. Blue represents gene regions located in the forward strand. Red represents gene regions located in the reverse strand. The inner bar plot represents per-base sequencing depth.

studies that have assessed the genetic structure of field populations of *Ae. aegypti* within the context of a specific intervention.

Following the intervention, there was a slight but significant decrease in nucleotide diversity, as evidenced by the population parameters Theta(S) and Theta(Pi). Moreover, the mitochondrial data for *Ae. aegypti* revealed statistically significant moderate stratification, indicating population differentiation between the pre- and post-intervention groups. Additionally, clustering analysis based on the allele frequency matrix revealed distinct clusters

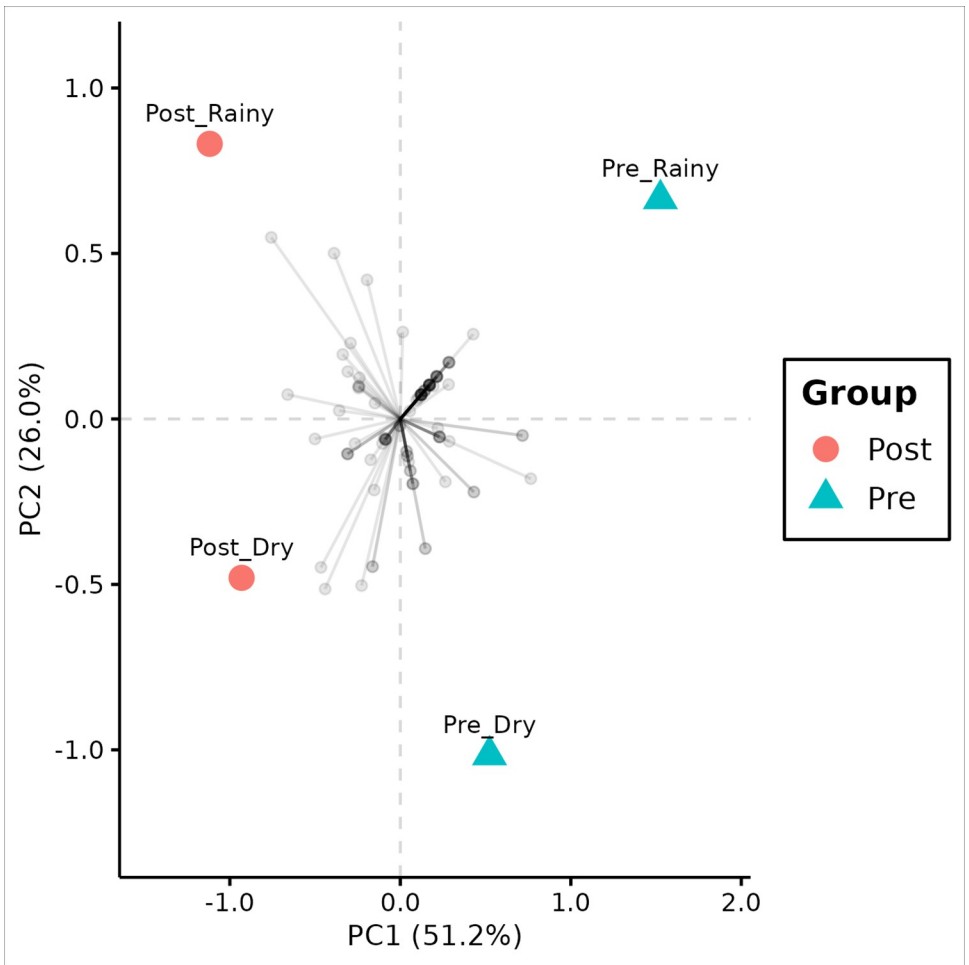

**Fig 5. PCA biplot based on the groupwise SNP frequency matrix.** Values between parentheses represent the variance explained by the respective principal component. Grey points represent SNP eigenvectors. Between-groups colored dashed lines represent significant hierarchical clustering groups.

between pre- and post-intervention groups, particularly along the first principal component, which indicates genetic variation associated with the intervention period. These groupings were statistically significant, as confirmed by the approximately unbiased p-value, which indicates significant clustering between the two groups considering their intervention period.

The PCA and hierarchical clustering analysis indicated a substantial genetic distance when comparing individuals from the pre- and post-intervention periods, indicating the presence of two distinct clusters. Checking these clusters, principal component 1 (PC1) demonstrates that there is a greater difference between intervention groups (51.2%) than between seasonal groups on principal component 2 (26.0%), based on the SNP frequency matrix (Fig 5). Furthermore, the identified clusters suggest that individuals from the post-intervention period showed a smaller *within* genetic distance compared to the pre-intervention period (Fig 7), converging with the results from nucleotide diversity estimates. Notably, no clustering was observed for the seasonal period variable, indicating that seasonality had no effect on the clustering analysis.

Based on the analysis described here, the scenario suggests that intervention induced a bottleneck effect on this population. This effect explains that there is an event in which a

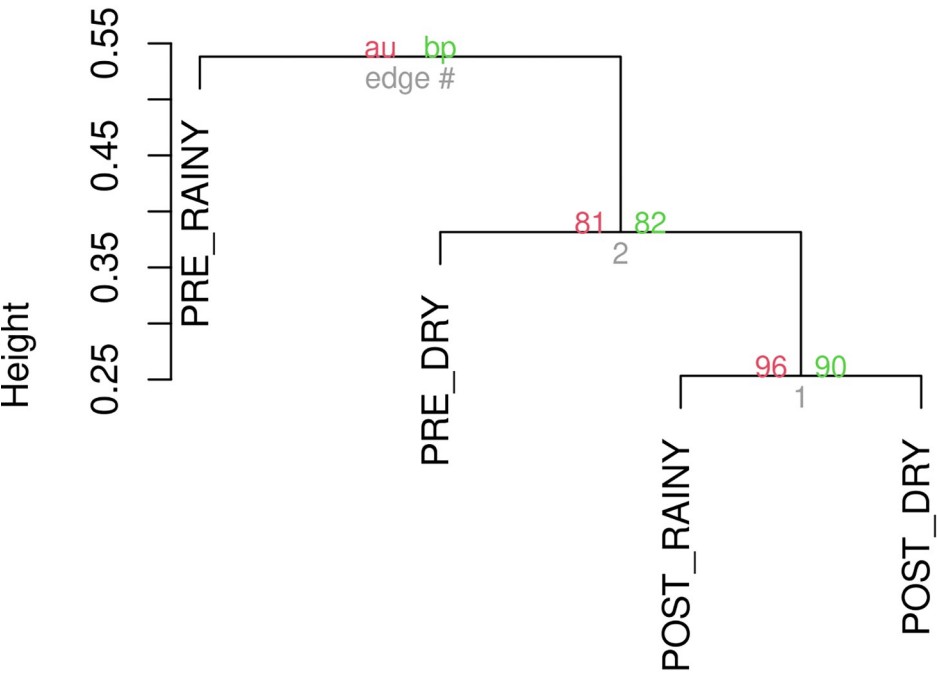

**Fig 6. Hierarchical clustering analysis and multiscale bootstrap resampling based on groupwise SNP frequency matrix.** Temporal periods: pre- (before intervention) and post- (after intervention). Seasonal periods: rainy and dry. AU (red): approximately unbiased p-value. BP (green): bootstrap probability.

population undergoes a drastic reduction in population density and subsequent population growth, in this case due to the pressure caused by the larvicide PPF and reintroduction of adult *Ae. aegypti* mosquitoes from nearby areas not exposed to PPF [31,32]. This is corroborated by the drastic reduction in the number of individuals in this region following the PPF intervention, wherein individuals from the pre-intervention period were nearly eliminated during larvicide application. Subsequently, the reintroduction from adjacent populations unaffected by PPF intervention occurred. Moreover, these data demonstrated that lcWGS is a highly effective and promising technique, not only due to its cost-effectiveness but also its capability to recover mitochondrial genome information at very low coverage.

The development of new sequencing approaches, such as whole genome sequencing (WGS) and double digest restriction-site associated DNA sequencing (ddRADseq), offers high sequencing depth and coverage of sequenced samples, but also comes with high execution costs. The lcWGS is a sequencing alternative for studies of population genetic structure, providing reliable data for marker selection, and is more economical than other techniques [17,33]. Additionally, there is a considerable cost reduction since our protocol uses one-fifth of the total volume recommended by the manufacturer for library preparation, and reducing the

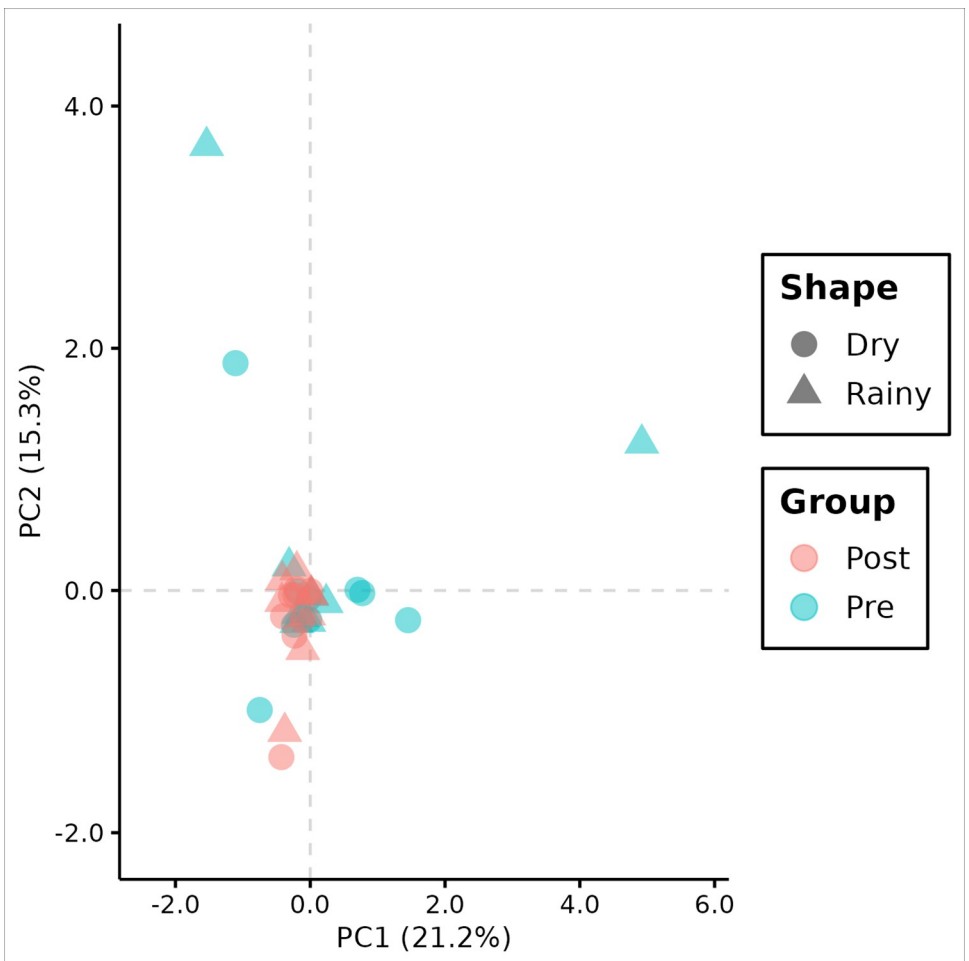

**Fig 7. Principal Component Analysis Biplot based on genotype matrix.** Values between parentheses represent the variance explained by the respective principal component. (circles): dry season. (triangles): rainy season. (blue): before intervention (pre-). (red): after intervention (post).

coverage from 20X to 1X, while simultaneously using a larger number of samples and data processing to increase the statistical power of our analyses [33,34]. This enables us to investigate the genetic structure and population dynamics of *Ae. aegypti*. lcWGS has proven to be a valuable technique for genetically characterizing populations of other medically important vectors belonging to the Culicinae subfamily [35], such as *Nyssorhynchus darlingi*, at a microgeographical scale and susceptibility to *Plasmodium* spp. infection [15,19,36].

Our findings also contribute significantly to emerging laboratory approaches such as genomic surveillance, within epidemiological surveillance, as a valuable tool using information from the genomes of any pathogen and/or its vectors for disease monitoring and control across diverse scenarios [37]. It aids in monitoring insecticide resistance, by revealing genetic mutations associated with it and avoiding the use of insecticides to which vectors have developed resistance. Also aids to development of personalized strategies, by identifying genetic diversity, which becomes possible to develop specific control strategies considering characteristics of local vector populations and increasing the effectiveness of intervention approaches [37,38]. Furthermore, it can be a useful tool for tracking arboviruses dissemination routes (i.e. tracking *Ae. aegypti* dispersion to understand the transmission of diseases such as dengue, Zika and

Chikungunya) and predicting and preventing epidemics and potential outbreaks (i.e. allowing health authorities to seek an effective response, before arboviruses cases reach alarming proportions) [39–41].

Despite the challenges and need for further studies, large-scale genomic surveillance emerges as a promising tool, considering the use of high-throughput genomic sequencing and epidemiological conditions to understand not only the previously mentioned issues, but also the genetic structure and population dynamics of several vectors of medical importance.

## Conclusion

Population genetic structure observed in this scenario, comparing *Ae. aegypti* specimens before and after the PPF intervention suggest a bottleneck effect, leading to a reduction in the mosquito population in this region. Subsequently, once the intervention was removed, the region experienced reintroduction by adjacent populations of *Ae. aegypti* unaffected by PPF exposure. These findings suggested that lcWGS consists in a valuable tool for genetically characterizing diverse vector populations in association with intervention periods and approaches, considering not only the cost-effectiveness of this technique, but also its contribution and importance to genomic surveillance. Notably, to the best of our knowledge, this study is the first to use field-collected *Ae. aegypti* to evaluate the genetic structure of individuals exposed to intervention methods recommended by the national dengue control program (PNCD). Despite the limitations, such as the low coverage of nuclear markers, additional studies using lcWGS are needed to better characterize the genetic structure of the vector mosquito population associated with genomic surveillance, especially for *Aedes aegypti*.

## Acknowledgments

We are grateful to Leônidas and Maria Deane Institute (Fiocruz Amazônia), Oswaldo Cruz Institute (Fiocruz) and Sao Paulo State University "Júlio de Mesquita Filho" (UNESP–Botucatu), who contributed to this work.

## Author Contributions

**Conceptualization:** Lorena Ferreira de Oliveira Leles, Sérgio Luiz Bessa Luz.

**Data curation:** Jose Joaquin Carvajal Cortés, Sérgio Luiz Bessa Luz.

**Formal analysis:** Lorena Ferreira de Oliveira Leles, Marcus Vinicius Niz Alvarez.

**Funding acquisition:** Sérgio Luiz Bessa Luz.

**Investigation:** Lorena Ferreira de Oliveira Leles, Jose Joaquin Carvajal Cortés.

**Methodology:** Lorena Ferreira de Oliveira Leles.

**Project administration:** Paulo Eduardo Martins Ribolla, Sérgio Luiz Bessa Luz.

**Resources:** Paulo Eduardo Martins Ribolla, Sérgio Luiz Bessa Luz.

**Software:** Marcus Vinicius Niz Alvarez.

**Supervision:** Diego Peres Alonso, Paulo Eduardo Martins Ribolla.

**Validation:** Lorena Ferreira de Oliveira Leles, Marcus Vinicius Niz Alvarez.

**Visualization:** Lorena Ferreira de Oliveira Leles, Marcus Vinicius Niz Alvarez, Jose Joaquin Carvajal Cortés, Diego Peres Alonso.

**Writing – original draft:** Lorena Ferreira de Oliveira Leles.

**Writing – review & editing:** Lorena Ferreira de Oliveira Leles, Marcus Vinicius Niz Alvarez, Diego Peres Alonso, Paulo Eduardo Martins Ribolla, Sérgio Luiz Bessa Luz.

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
