## [Decision Letter · Decision Letter 0]

20 Jun 2024

Dear Miss Leles,

Thank you very much for submitting your manuscript "Evaluation of Aedes aegypti control intervention with pyriproxyfen by lcWGS in Manacapuru, Amazonas, Brasil" for consideration at PLOS Neglected Tropical Diseases. As with all papers reviewed by the journal, your manuscript was reviewed by members of the editorial board and by several independent reviewers. The reviewers appreciated the attention to an important topic. Based on the reviews, we are likely to accept this manuscript for publication, providing that you modify the manuscript according to the review recommendations. 

Sincerely,

Paul O. Mireji, PhD

Section Editor

Paul Mireji

Section Editor

Reviewer's Responses to Questions

**Key Review Criteria Required for Acceptance?**

**Methods**

-Are the objectives of the study clearly articulated with a clear testable hypothesis stated?

-Is the study design appropriate to address the stated objectives?

-Is the population clearly described and appropriate for the hypothesis being tested?

-Is the sample size sufficient to ensure adequate power to address the hypothesis being tested?

-Were correct statistical analysis used to support conclusions?

-Are there concerns about ethical or regulatory requirements being met?

Reviewer #1: Objective of the study needs to be reframed as it is not apparent to the reader.

Reviewer #2: Yes the methods are coeherent with the objectives and are well described to allow to replicate the experiments

Sample size is fine and appropriate to test initial hypothesis.

there is no concern about ethical and/or regulatory issues

**Results**

-Does the analysis presented match the analysis plan?

-Are the results clearly and completely presented?

-Are the figures (Tables, Images) of sufficient quality for clarity?

Reviewer #1: The captions for Figure 4 and 5 need to include additional information to enhance clarity. Terms such as 'Post' and 'Pre' should be briefly explained within the captions to provide context for the reader.

Reviewer #2: The analysis presented do match the analysis plan.

The results are clearly presented and are original and complete.

Figures/tables are clear and self-explaining

**Conclusions**

-Are the conclusions supported by the data presented?

-Are the limitations of analysis clearly described?

-Do the authors discuss how these data can be helpful to advance our understanding of the topic under study?

-Is public health relevance addressed?

Reviewer #1: (No Response)

Reviewer #2: Conclusions are supported by the data presented and are well discussed.

The topic of the manuscript is of relevant interest and its public health relevanca is properly addressed

**Editorial and Data Presentation Modifications?**

Reviewer #1: Please revise the manuscript to adhere to the journal's standards, including the abstract format and citation style. Additionally, take into account the annotations provided in the manuscript PDF to enhance clarity, particularly in terms of language structure.

Reviewer #2: (No Response)

**Summary and General Comments**

Reviewer #1: The manuscript needs a clear justification for its claim of relative cost-effectiveness, particularly in comparison to other available methods. Specifically, how does the cost of lc WGS compare to alternative approaches? This clarification is essential for assessing the study's practical implications and potential adoption in relevant fields.

The map presented in Figure 1 would benefit from including coordinates relative to the North arrow for improved orientation. Furthermore, it's important to ensure compliance with copyright regulations when utilizing data from Google Maps in QGIS. The authors should verify that any data sourced from Google Maps are permissible under the cc-by 4.0 license.

The concept of the bottleneck effect, referenced in both Line 47 and Line 246, is inadequately explained in the context of the population under study. Given its significance, a more thorough elucidation is warranted to ensure clarity and understanding for the reader. This discussion is crucial and should be more prominently addressed within the manuscript.

Reviewer #2: A nice manuscript, well written and with original data that represent a relevant step-forward of teh knowledge in the field

PLOS authors have the option to publish the peer review history of their article (what does this mean?). If published, this will include your full peer review and any attached files.

Reviewer #1: No

Reviewer #2: No

Figure Files:

Data Requirements:

Reproducibility:

References

---

## [Decision Letter · Decision Letter 1]

18 Sep 2024

Dear Miss Leles,

We are pleased to inform you that your manuscript 'Evaluation of Aedes aegypti control intervention with pyriproxyfen by lcWGS in Manacapuru, Amazonas, Brasil' has been provisionally accepted for publication in PLOS Neglected Tropical Diseases.

Best regards,

Paul O. Mireji, PhD

Section Editor

Paul Mireji

Section Editor

Reviewer's Responses to Questions

**Key Review Criteria Required for Acceptance?**

**Methods**

-Are the objectives of the study clearly articulated with a clear testable hypothesis stated?

-Is the study design appropriate to address the stated objectives?

-Is the population clearly described and appropriate for the hypothesis being tested?

-Is the sample size sufficient to ensure adequate power to address the hypothesis being tested?

-Were correct statistical analysis used to support conclusions?

-Are there concerns about ethical or regulatory requirements being met?

Reviewer #1: (No Response)

**Results**

-Does the analysis presented match the analysis plan?

-Are the results clearly and completely presented?

-Are the figures (Tables, Images) of sufficient quality for clarity?

Reviewer #1: (No Response)

**Conclusions**

-Are the conclusions supported by the data presented?

-Are the limitations of analysis clearly described?

-Do the authors discuss how these data can be helpful to advance our understanding of the topic under study?

-Is public health relevance addressed?

Reviewer #1: (No Response)

**Editorial and Data Presentation Modifications?**

Reviewer #1: (No Response)

**Summary and General Comments**

Reviewer #1: The comments to the author have been addressed. Including a couple of sentences on the Insect Growth Regulation action of Pyriproxyfen on mosquito larvae in the background section would enhance the paper's readability, especially for readers interested in larval source management. While this is not mandatory, it would greatly improve the paper.

PLOS authors have the option to publish the peer review history of their article (what does this mean?). If published, this will include your full peer review and any attached files.

Reviewer #1: No

---

## [Editor Report · Acceptance letter]

27 Sep 2024

Dear Miss Leles,

We are delighted to inform you that your manuscript, "Evaluation of Aedes aegypti control intervention with pyriproxyfen by lcWGS in Manacapuru, Amazonas, Brazil," has been formally accepted for publication in PLOS Neglected Tropical Diseases.

Best regards,

Shaden Kamhawi

co-Editor-in-Chief

Paul Brindley

co-Editor-in-Chief
